# Evaluation of Genotoxic and DNA Photo-Protective Activity of *Bryothamnion triquetrum* and *Halimeda incrassata* Seaweeds Extracts

**Ángel Sánchez-Lamar** [1,*], **Maribel González-Pumariega** [1], **Fabiana Fuentes-León** [1,*], **Marioly Vernhes Tamayo** [2], **André P. Schuch** [3] and **Carlos F. M. Menck** [3]

1   Plant Biology Department, Faculty of Biology, University of Havana, Havana 10400, Cuba; bel@nauta.cu
2   Radiobiology Department, Center of Applied Technology and Nuclear Development (CEADEN), Havana 11300, Cuba; mariolys@ceaden.edu.cu
3   Microbiology Department, Instituto de Ciências Biomédicas, Universidade de São Paulo, São Paulo 03178-200, Brazil; aschuch@usp.br (A.P.S.); cfmmenck@usp.br (C.F.M.M.)
*   Correspondence: alamar@fbio.uh.cu (Á.S.-L.); fabiana@fbio.uh.cu (F.F.-L.); Tel.: +53-7-832-8542

**Abstract:** The ultraviolet (UV) component of sunlight is high on the earth surface, especially at low latitudes, raising the risk of skin diseases, including cancer. The use of natural compounds is a strategy to protect people against UV damage. Seaweeds are becoming increasingly influential in the food industry, and are also used in the pharmacy and cosmetic industries, due to several bioactive demonstrated properties. This work analyzed the genotoxic and photoprotective effects of the aqueous extracts of two seaweed species: *Bryothamnion triquetrum* and *Halimeda incrassata*. A cell-free plasmid DNA assay was employed, allowing detection of DNA breaks. The plasmids were exposed to increasing concentrations of aqueous extracts. DNA break was produced at concentrations of 2.0 and 4.0 mg/mL in both seaweed extracts and, consequently, a genotoxic effect is postulated. This effect arises with higher exposure times. Additionally, different combinations of plasmid DNA, restriction enzymes (*Eco* RI, *Bam* HI, and *Pvu* II) and extracts were assayed. The extracts did not produce an interference effect in the reconnaissance of the specific restriction target sequences of each enzyme. Photoprotective activity of the extracts was evaluated in UVC-irradiated plasmids. None of the extracts displayed DNA protective effects in this assay.

**Keywords:** plasmid DNA; UV radiation; DNA protection

## 1. Introduction

The ultraviolet (UV) component of sunlight that reaches the Earth's surface deserves social concern, because it generates DNA damages that are related to the development of skin diseases, including cancer. Actually, it is a perspective that sun protection creams include compounds capable of absorbing radiation and improving the DNA-damage repair [1]. The evaluation of photoprotective agents, including the assessment of their efficacy and safety and the mechanisms involved, are highly relevant.

Seaweeds have attracted much attention as a source of natural preparations, with potential applications in the cosmetic industry and biomedicine [2,3]. Also, seaweeds are excellent photoprotective candidates, because they are exposed to UV radiation and have developed several defense mechanisms [4]. As a preclinical requirement, the employment of these natural sources involves a genotoxic evaluation. The crude extracts from seaweeds are complex mixtures of phytocompounds that may contain toxic and beneficial properties [5]. For instance, some alkaloids, terpenoids, steroids and flavonoids have

been identified as protective but also damaging agents [6] and seaweed-derived products are frequently constituted by these phytochemical classes.

The aqueous extracts from *Halimeda incrassata* and *Bryothamnion triquetrum* seaweeds have been reported as antiatherogenic [7] and antinociceptive and anti-inflammatory [8,9], respectively. Additionally, both extracts contain neuroprotective and antioxidant capacities [10–12] and some experimental works have been carried out in order to evaluate their toxic properties, by means of in vitro assays [13–15]. The present study was designed to test the genotoxic potential of these aqueous extracts in the primary structure of DNA molecule. After, we evaluated the ability to protect DNA against the damage induced by UVC radiation.

## 2. Materials and Methods

### 2.1. Aqueous Extracts Preparation

Seaweeds *Halimeda incrassata* (Ellis) Lamouroux (Chlorophyta: Halimedaceae, Bryopsidales) and *Bryothamnion triquetrum* (S.G. Gmelim) Howe (Rhodophyta: Rodomelaceae, Ceramiales) were collected in June of 2009 in Bajos de Santa Ana beach, Cuba. Samples were sent to Arsenio Areces Mallea, Department of Biochemistry, Oceanology Institute of the Cuban Ministry of Science, Technology and Environment, where they were authenticated. Dry and powdered collected specimens of *H. incrassata* and freshly collected specimens of *B. triquetrum* were homogenized in distilled water 1:5 (*w/v*) and 1:4 (*w/v*), respectively, following a method previously described [16]. These aqueous extracts were further lyophilized.

### 2.2. Experimental Biological System and Procedures

Supercoiled circular DNA pBluescript SK II (2961 bp) was employed (100 ng/μL). The restriction enzymes used (*Eco* RI, *Bam* HI, *Pvu* II, T4-endoV), MULTI-CORE Buffer and bovine serum albumin (BSA) were acquired from Promega (Madison, WI, USA). As negative control were used 100 ng of plasmid DNA and 19 μL of NET buffer (100 mM NaCl, 10 mM EDTA, 10 mM Tris HCl, pH 8). As positive control was used UVC-irradiated DNA plasmid (100 ng) and NET buffer (18 μL) incubated and digested with T4-endo V (1 μL) at 37 °C for 30 min. In addition to negative and positive controls of DNA conformations, a buffer control and an enzymatic digestion control were used (not shown). For all treatments, final volume was 20 μL and was completed with NET buffer. Each experiment was repeated at least three times to obtain reliable results.

#### 2.2.1. Genotoxicity of Seaweeds Extract

Plasmids were diluted (100 ng/μL) in TE solution (10 mM Tris-HCl, 1 mM EDTA, pH 7.5) and dispensed into reaction tubes (1 μL/tube). Concentrations 0.01, 0.1, 0.5, 1.0, 2.0 and 4.0 mg/mL of seaweed aqueous extracts were evaluated as inducers of DNA plasmid breakage. In all cases, reaction mixtures (20 μL/tube) were incubated at 37 °C for 30 min. To evaluate the influence of time exposure to aqueous extracts, three different times for reaction mixture incubation (30, 60 and 90 min) were assayed. In this case, only concentrations 1.0 and 2.0 mg/mL of both extracts were used.

In order to separate the conformations of plasmid DNA: supercoiled native conformation (form SF), nicked conformation (form NF) resulting from single-strand breaks (SSB) and linear conformation (form LF) resulting from double-strand breaks (DSB), an electrophoresis in 0.8% agarose gels in TBE buffer (89 mM Tris–$H_3BO_3$, 2 mM EDTA, pH 8.1) was performed. Aliquots from each sample (100 ng/tube) were mixed with 3 μL of loading buffer (75 mM EDTA; 50% glycerol; 0.2% bromophenol blue), and applied in a horizontal gel electrophoresis chamber, at 100 volts 60 mA during 60 min.

Rather than possible clastogenic actions of extracts, other effects on DNA were studied. In this sense, we evaluated the eventual inhibition of *Eco* RI, *Bam* HI and *Pvu* II by extracts using two approaches:

(1)  *Eco* RI, *Bam* HI and *Pvu* II enzyme mix (3 μL) were incubated with 1.0 mg/mL of seaweeds extracts (5 μL) in multi-core buffer (2 μL), BSA (2 μL) and deionized water (3 μL) during 30 min. After, the mixes were complemented with plasmid DNA (500 ng).

(2)  Plasmid DNA (500 ng) was incubated with 1.0 mg/mL of seaweed extracts (5 μL) during 30 min. Afterwards, the mixes were complemented with multi-core buffer (2 μL), BSA (2 μL), deionized water (3 μL) and *Eco* RI, *Bam* HI and *Pvu* II enzyme mix (3 μL).

Enzymatic digestion was carried out during 150 min at 37 °C (Promega). Then, the mixtures were submitted to a 1.5% agarose electrophoresis gel in TAE 1x buffer, during 5 h.

In all cases, gel was stained with ethidium bromide (0.1 ng/μL) and DNA bands were visualized by fluorescence in a Benchtop UV transilluminator system (Model FTM20, FUSE 2XT3, 2 AMP 25 W, Ultra-Violet Products Ltd, Cambridge, UK). Permanent records were performed using a Polaroid system (SAMSUNG S630 FC Samsung opto-electronics CO. LTD, Tianjin, China). The change of pattern of electrophoretic running was taken of criterion of extract interaction with DNA or enzyme inhibition.

### 2.2.2. Antigenotoxicity of Seaweed Extracts

Initially, the effect of time exposure to seaweed extract on T4-endo V was evaluated. The enzyme T4-endoV (1 μL) was exposed to 1.0 mg/mL of seaweed aqueous extracts (1 μL) during 10, 30 and 50 min at 37 °C, when irradiated plasmids were added to the mixture. For photoprotection, DNA (100 ng) was incubated with 0.01, 0.1, 0.5 and 1.0 mg/mL concentrations of seaweed (1 μL) and immediately UVC-irradiated. The criterion of enzymatic inhibition and antigenotoxic effects was the reduction of NF bands and presence of SF bands. UVC irradiation was carried out in uncovered 3 cm diameter Petri dishes using a Vilber Loumart Lamp T15M 15 W and λ = 254 ηm. The dose used value was 200 J/m$^2$, corresponding to 9 s. After every treatment, T4-endoV (1 μL) was added and incubated at 37 °C during 30 min.

### 3. Results

The clastogenic effects of aqueous extracts of *H. incrassata* (HiE) and *B. triquetrum* (BtE) on pBluescript II SK plasmid conformation are showed in Figure 1. For both extracts, the higher concentrations assayed (2.0 and 4.0 mg/mL) provoke breaks on plasmid DNA, originating NF bands.

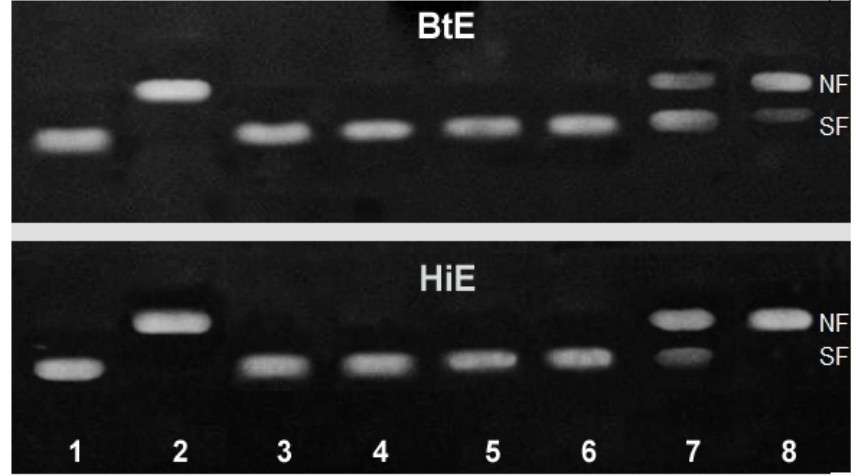

**Figure 1.** Clastogenic effect of different concentrations of *B. triquetrum* and *H. incrassata* extracts on plasmid DNA during 30 min: (1) Negative control; (2) Positive control; (3–8) DNA exposed to 0.01, 0.1, 0.5, 1.0, 2.0 and 4.0 mg/mL of seaweed extracts. SF: supercoiled form. NF: open nicked form.

To evaluate the incidence of extract incubation time, concentrations 1.0 and 2.0 mg/mL of seaweed extracts were selected. The results obtained with treatment lasting 30, 60 and 90 min are shown in

Figure 2. It was observed that longer incubation time produces NF or even LF, indicating that the extracts induced DNA breaks, including single- and double-strand breaks. This genotoxic effect is stronger for *H. incrassata,* when compared to *B. triquetrum* extract.

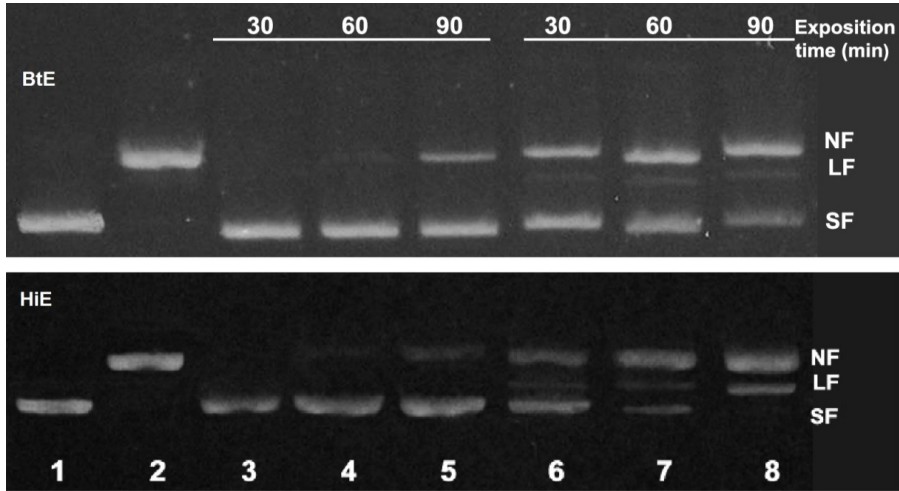

**Figure 2.** Clastogenic effect of *B. triquetrum* and *H. incrassata* extracts on plasmid DNA during different incubation times. (1) Negative control; (2) Positive control; (3–5) DNA exposed to 1.0 mg/mL during 30, 60 and 90 min, respectively; (6–8) DNA exposed to 2.0 mg/mL during 30 and 90 min, respectively. SF: supercoiled form. NF: open nicked form. LF: linear form.

The generation of DNA damage that would inhibit restriction enzymes on DNA was also tested and the result is shown in Figure 3. The control electrophoretic restriction pattern from simultaneous digestion was constituted by the bands corresponding to 2417, 258 and 169 bp. The restriction enzymatic pattern in the evaluated treatment was similar to the control: 1.0 mg/mL of both extract did not alter the activities of *Eco* RI, *Bam* HI and *Pvu* II enzymes.

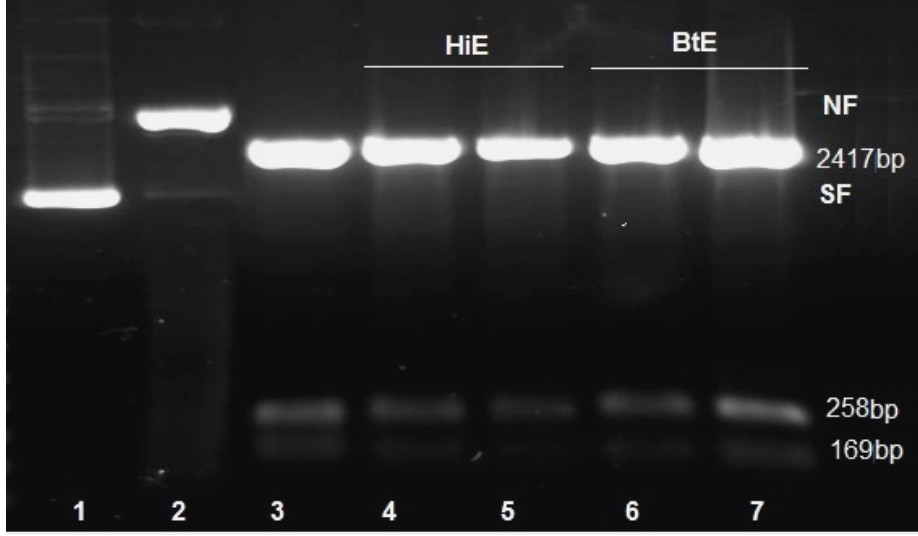

**Figure 3.** Inhibition of restriction enzymes (*Eco* RI, *Bam* HI and *Pvu* II) activities by *B. triquetrum* and *H. incrassata* extract. (1) Negative control; (2) Positive control; (3) DNA digested with restriction enzymes; (4,6) extract-DNA incubation previous to enzyme addition; (5,7) extract-enzyme incubation previously to DNA addition.

Previous to photoprotective evaluation of seaweed extracts, its effects on T4-endo V activity was tested. BtE and HiE, at any exposure time, did not inhibit the nicking enzyme activity on DNA damaged (Figure 4).

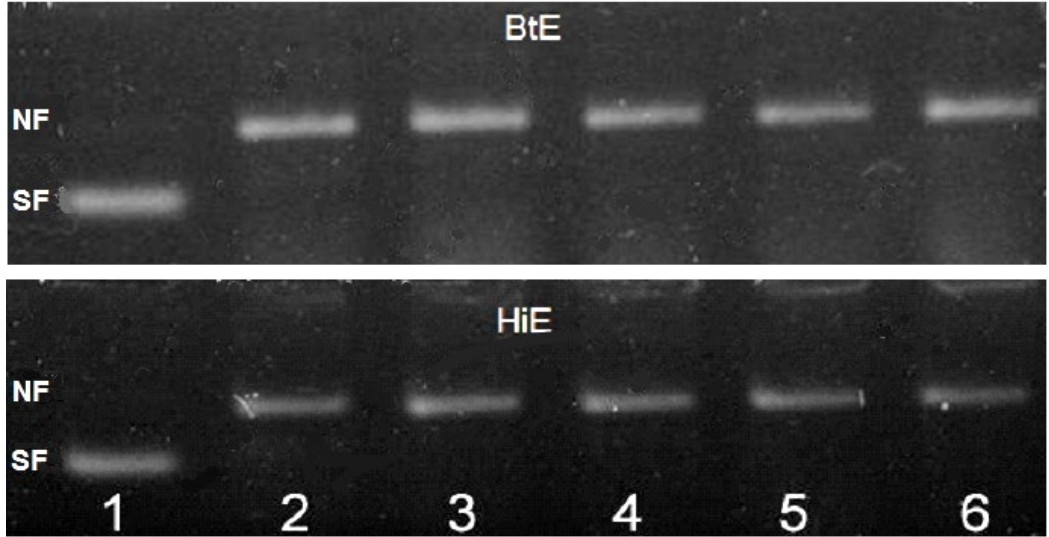

**Figure 4.** Inhibition of T4-endo V activity by *B. triquetrum* and *H. incrassata* extracts. (1) Negative control; (2) Positive control; (3–6) T4-endo V exposed to extract (1.0 mg/mL) during 0, 10, 30 and 50 min. SF: supercoiled form. NF: open nicked form.

The protective effect against UVC radiation indicates that both extracts (at concentrations lower than 1.0 mg/mL) did not reduce the amount of DNA photodamage (Figure 5).

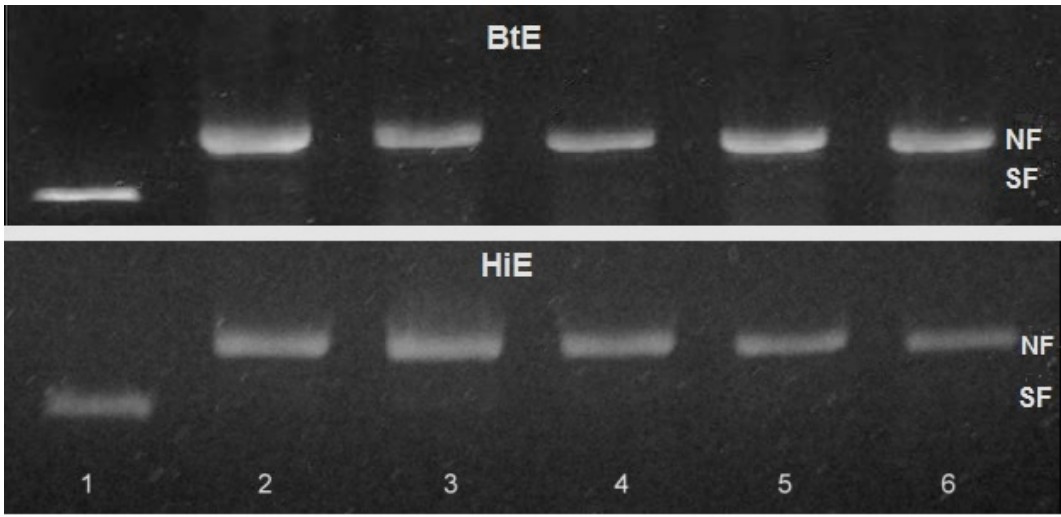

**Figure 5.** Photoprotection of *B. triquetrum* and *H. incrassata* extracts on plasmid DNA-UVC-irradiated (200 J/m$^2$), and then digested with T4-endo V enzyme: (1) Negative control; (2) Positive control; (3–6) DNA exposed to 0.01, 0.1, 0.5 and 1.0 mg/mL of extracts. SF: supercoiled form. NF: open nicked form.

## 4. Discussion

*B. triquetrum* and *H. incrassata* are two species of different genera greatly abundant in the Caribbean Sea. In the possible utilization of these species as phytopharmaceuticals and/or photoprotective agents,

the assessment of their genotoxic effects is an important contribution [3,5]. In this sense, there is not much available information. The assays used in the precedent genotoxic studies performed with these extracts only cover the chromosome level [13].

The cell-free plasmid DNA assay is a sensitive method for the measurement of DNA damage at the primary structure level and thoroughly used in current genotoxic and antigenotoxic studies [17–19]. The current paper focused on the genotoxic analysis of two specific endpoints: the capacity of producing breaks in plasmid DNA, and some eventual modifications of DNA bases, which could inhibit restriction enzyme activities.

In both seaweed extracts, the clastogenic properties were similar for the concentrations and treatment times of DNA breaks onset. However, HiE extract provoked more breaks than BtE extract, at the higher concentration (4.0 mg/mL). The damage generation (single and double strand) was dependent on the DNA exposure time with both extracts (Figures 1 and 2).

The BtE and HiE are complex mixtures of natural substances. Phytochemistry studies revealed the presence of abundant phenolic compounds in both extracts [16,20]. Several polyphenols are reported in literature as natural toxicants and some of them can be genotoxic at specific concentrations, such as alkaloids [21] and phenolic acids. The salicylic and ferulic acid found in HiE [20] and p-coumaric, t-cinnamic and ferulic acid in BtE [16] could be responsible for the clastogenic effect observed. Particularly, the red marine algae possess flourtannins, polymers of high molecular weight with monomeric units of flourglucinol and sulphated polyphenolic compounds [22] that could also justify the genetic damage in BtE. It is known that the metal content is another aspect of the extracts' genotoxic potential. In this sense, the chemical composition of BtE, the presence of toxic metals Pb and Cd are respectively high and moderate ($7.5 \pm 0.1$ and $0.2 \pm 0.08$ mg/kg, for each one) [23]. In the case of HiE, the content of metals has not been informed.

Another form of harming DNA could be through reactions of deamination, alkylation, oxidation, bulky additions or some base interactions [24]. The use of endonuclease restriction enzymes permitted, in an indirect way, to obtain experimental evidence of DNA damage. Inhibition of enzymes and/or DNA base modification in restriction sequences could avoid the cleavages, affecting electrophoretic pattern [25]. The extracts (1.0 mg/mL) did not change the pattern of restriction enzymatic action on plasmid DNA, demonstrating that BtE and HiE extracts: (i) did not inhibit enzymatic action; (ii) did not produce modification in the base sequence, interfering in its recognition (Figure 3). The results suggest that neither of the two extracts alters the bases of the DNA nor inhibits the endonuclease activity of restriction enzymes.

At this point, in our experimental conditions, concentrations ≤1.0 mg/mL during 30 min for both extracts did not damage plasmid DNA. These results agree with other authors' reports. In this sense, BtE and HiE did not induce chromosome aberrations in Chinese hamster ovary cells at 0–1.0 mg/mL concentrations [13,26]. Also, BtE was not mutagenic in *S. typhimurium* with Ames Test [15].

For photoprotection experiments, non-genotoxic conditions were selected, concentrations from 0.01 to 1 mg/mL and 30 min time exposure. In addition, it was demonstrated that both extracts did not affect T4-endo V enzymatic activity in any time assayed (Figure 4). Even so, the BtE and HiE did not reduce the UVC-induced DNA damage levels (Figure 5), indicating no photoprotective activity. Similar results have been reported for other Cuban seaweed: *Halimeda monile*, using plasmid DNA as a biological assay [27].

Although DNA ex vivo assay is excellent to detect direct damage in DNA molecule, our procedure was focused on find antigenotoxicity through UV absorption. It is possible that in these seaweed extracts, chromophore compounds concentration could not be enough to exert this physical protection.

On the other hand, some phytocompounds have demonstrated protective capacities increasing UV-DNA repair machinery efficiency or dismissing mutations frequency [1,28,29]. However, this assay did not allow for the evaluation of photoprotective properties by bioantimutagenic action. In this sense, future studies using cell systems could still show positive results.

## 5. Conclusions

The present work shows that concentrations lower than or the same as 1.0 mg/mL of *Bryothamnion triquetrum* and *Halimeda incrassata* extracts did not produce genotoxic effects; higher doses have potential harm on DNA. Photoprotective capacity was not found in both extracts. The present study constitutes the first report of genotoxicity and antigenotoxicity against UV radiation, of *Bryothamnion triquetrum* and *Halimeda incrassata* extracts using an ex vivo assay.

**Acknowledgments:** CAPES (Brazil)-MES (Cuba) collaborative project financed this work. We did not receive funds for covering the costs to publish in open access.

**Author Contributions:** Ángel Sánchez-Lamar, Carlos Federico Martin Menck and Maribel González-Pumariega conceived and designed the experiments; Maribel González-Pumariega and Marioly Vernhes Tamayo performed the experiments; Ángel Sánchez-Lamar, Maribel González-Pumariega, Marioly Vernhes Tamayo and Fabiana Fuentes-León analyzed the data; André P. Schuch contributed reagents/materials/analysis tools; Ángel Sánchez-Lamar and Fabiana Fuentes-León wrote the paper.

**Conflicts of Interest:** The authors declare no conflict of interest.

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
