# Peer review of "Evaluation of Genotoxic and DNA Photo-Protective Activity of Bryothamnion triquetrum and Halimeda incrassata Seaweeds Extracts"

_cosmetics, doi:10.3390/cosmetics4030023_

Round 1

Reviewer 1 Report

The authors Ángel Sánchez-Lamar, Maribel González-Pumariega, Fabiana Fuentes-León, Marioly Vernhes Tamayo, André P. Schuch and Carlos F. M. Menck report in their manuscript “Evaluation of genotoxic and DNA photo-protective activity of Bryothamnion triquetrum and Halimeda incrassata seaweeds extracts“ that

(i)          both aqueous extracts produced DNA strand breaks in cell free DNA plasmids (Supercoiled circular DNA pBluescript SK II) measured as change of supercoiled form to open nick form in a dose of 2.0 and 4.0 mg/mL after 30min; lower doses were effective after prolonged incubation time.

(ii)         The ability to change the restriction enzyme pattern caused by Eco RI, Bam HI, Pvu II, due to base modifications was assessed, but no effects were observed, no matter whether DNA/extract or restriction enzyme/extract interactions were assessed.

(iii)       In addition, the ability to modify the nicking ability of T4endo V was checked with no negative results observed for the extracts employed.

(iv)       No DNA protective effects in UVC irradiated plasmids were observed.

This paper provides interesting information, but I have some problems to understand the approach:

Major criticism

1.      Do you want to present new types of assays to address the listed issues? Then you should compare the current assay to established assays and describe the advantage of the new one as compared to the ones already on the market. e.g comet assay or Bacterial Reverse Mutation Assay (Ames test).

2.      If you want to describe the effects of the two extracts it would be wise to add additional controls using extracts with known, already established genotoxic effects, in order to help judging your findings in comparison to other plant extracts.

3.      Why did you use UVC as UV radiation source at list for experiments done in Figure 5? As some of the authors are experts in the field I just wonder that you did not use UVA, UVB or a solar simulator. UVC is the most unphysiologcial UV quality. See also “Thus, the solar UV radiation of relevance to human health and ecosystems consists of UVA and UVB wavelengths” by Schuch et al in dx.doi.org/10.1016/j.freeradbiomed.2017.01.029.

Can you comment on the relevance of this kind of assay for UVA induced damage? As far as I know UVA does not induce double strand breaks. (Cadet et al, JID 2011). In addition, by using UVC as damage inducing UV source you will not detect UVA protective effects.

4.      In your discussion (page 6, line 177-178) you argue that “The salicylic and ferulic acid found in HiE [20] and p-cumaric, t-cinamic and ferulic acid in BtE [16] could be responsible for the clastogenic effect observed.” I just wonder, because some of these ingredients are included as antioxidant in products already on the market. Are the concentrations you found in your paper from 2011 (Ref 20) relevant in comparison to the concentrations regularly found in products? To my knowledge, the doses in final products are quite low.

Minor criticism

A native English speaking person should go over the manuscript to eliminate some mistakes.

Author Response

Thanks for your suggestion about our paper titled: Evaluation of genotoxic and DNA photo-protective activity of Bryothamnion triquetrum and Halimeda incrassata seaweeds extract. In order to respond the reviewer comments some modification were made in document. We are sending the answers demanded.

1. In the study of compounds capable to damage or protect DNA is common to use assays that report change in DNA primary structure. In this work, the goal was evaluate the genotoxic and antigenotoxic action of seaweeds extracts. In this sense, ex vivo experiments using cell-free plasmid DNA are useful and commonly used. Compared to Comet assay and Ames test, the experimental model used gives information about direct interaction with DNA molecule without any other interaction. Compounds that cause DNA strand breaks directly can be evaluated through this assay. Additionally the use of plasmidic DNA joined the employment of enzymes that recognize specific type of damage in DNA structure, allows the detection of specific lesions. Therefore a valid alternative for the evaluation of the genotoxic and photoprotective capacity of natural products, is the use of this biological model combining or not the ultraviolet radiation as source of damage and T4 endo V enzyme. In general, for genotoxic and/or antigenotoxic evaluations of different compounds, this experimental model have proved be useful, for its fast-response and sensibility. Also it is internationally validated (Attaguile et al. 2000; Moreno et al. 2004; Kejnovský et al. 2004; Schuch et al. 2009, 2010, 2013, 2014; Guha et a.l 2011; Vernhes et al. 2013; González-Pumariega et al. 2016)

2. We agree that plants extracts with know genotoxic properties would be useful. One of the aims of this paper was to evaluate the genotoxic effects of these two seaweeds aqueous extracts. These genotoxic effects could be the DNA strand breaks generation that causes change in plasmid structure. These changes originate three conformational forms. In our work, non treated plasmids (negative control) and irradiated plasmids treated with T4 endo V (positive control) were considered as controls of supercoiled and damaged form of DNA respectively.

3. Sunlight ultraviolet (UV) radiation wavelength could be divided into UVA (315-400 nm), UVB (280-315 nm) and UVC (100-280 nm), from less to the most energetic and mutagenic. Short wavelength UVC radiations are completely absorbed by the ozone layer, but there are important artificial sources commonly used in medicine and for cosmetic purposes (De Flora, 2013). Furthermore, stratospheric ozone layer depletion, together with climate change and global warming, predict an uncertain scenario for UV Earth’s incidence in the next decades (McKenzie et al. 2011), rising concerns about its biological consequences and motivating worldwide scientific efforts to develop better photoprotection strategies (González et al. 2011). In this sense to used UVC radiation for testing DNA damage is common in literature (Patrick 1997; Douki et al. 2002, 2013; Markovitsi et al. 2010; Banyasz et al. 2012).

Using this same assay UVA irradiation could also be studied. It is know that UVA cause oxidative DNA damage. Fpg enzyme recognizes and cut DNA in oxidize bases. A plasmid ex vivo assay using this fpg enzyme would allow detecting the effect of UVA on DNA molecule (Schuch et al 2010). On the other hand, it is already studied that UVA light can causes direct lesions on DNA (CPD lesions or 6,4 PP) (Jiang et al. 2009; Schuch et al. 2009). In this sense, studies in order to evaluate protective effects against UVA light could be performed as our work.

4. Literature reported that phenolic acids could play different roles in biological system, positives and hazardous (Korkina et al. 2012). In this sense, concentration and different conjugations of compounds are highly relevant. There are some evidence of toxic effects for salicylic (Madan y Levitt, 2014), ferulic (Mancuso y Santangelo, 2014), cinnamic (Li et al. 2017) and coumaric acids (Pei et al. 2016). Also, according to EWG's Skin Deep (an online safety guide for cosmetics and personal care products) ferulic and salicylic acids have been prohibited for use in some types of cosmetics in Japan and Canada. Moreover these compounds possess chemical structures that could probably interact with DNA molecule and for that reason we have postulated they could be responsible for the clastogenic effect observed. 

5. The changes proposed and other several English corrections were made.

We are attaching you a modified manuscript

Thank you for your time

1.     Attaguile, G.; Russo, A.; Campisi, A.; Savoca, F.; Acquaviva, R.; Ragusa, N. and Vanella, A. Antioxidant activity and protective effect on DNA cleavage of extracts from Cistus incanus L. and Cistus monspeliensis L. Cell Biol. Toxicol. 2000; 16: 83-90.

2.     Banyasz, A. Douki, T. Improta, R. Gustavsson, T. Onidas, D. Vaya, I. Perron M. and D. Markovitsi. Electronic excited states responsible for dimer formationupon UV absorption directly by thymine strands: joint experimental and theoretical study, J. Am. Chem. Soc. 2012, 134, 14834–14845.

3.     De Flora S. Genotoxicity and carcinogenicity of the light emitted by artificial illumination systems. Arch Toxicol 2013 87, 403-405.

4.     Douki, T. Vadesne-Bauer G. and Cadet J., Formation of 2′-deoxyuridine hydrates upon exposure of nucleosides togamma radiation and UVC-irradiation of isolated andcellular DNA, Photochem. Photobiol. Sci. 2002, 1, 565–569.

5.     González S, Gilaberte Y, Philips N, Juarranz A Current trends in photoprotection - A new generation of oral photoprotectors. Open Dermatol J 2011 5, 6-14.

6.     González-Pumariega, M.; Fuentes-León, F.; Vernhes, M.; Schuch, A.P.; Menck, C.F.M.; Sánchez-Lamar, Á. El extracto acuoso de Cymbopogon citratus protege al ADN plasmídico del daño inducido por radiación UVC. Ars Pharm. 2016, 57, 193-199.

7.     Guha G, Rajkumar V, Mathew L and Kumar RA. The antioxidant and DNA protection potential of Indian tribal medicinal plants. Turkish J.  Biol. 2011, 35, 233-42.

8.     Jiang, Y. Rabbi, M. Kim, M. Ke, C. Lee, W.  Clark, R. L. Mieczkowski, P. A. and Marszalek, P. E. UVA Generates Pyrimidine Dimers in DNA Directly. Biophys J. 2009 96(3): 1151–1158.

9.     Kejnovský, E.; Nejedlý, K.; Kypr, J. Factors influencing resistance of UV-irradiated DNA to the restriction endonuclease cleavage. Int. J. Biol. Macromol. 2004, 34, 213-222.

10.   Korkina, L.; Luca, C.D.; Pastore, S. Plant polyphenols and human skin: friends or foes. Ann. N. Y. Acad. Sci. 2012, 1259, 77-86.

11.   Li, J. He, D. Wang, B. Zhang, L. Li, K. Xie, Q. and Zheng, L. Synthesis of hydroxycinnamic acid derivatives as mitochondria-targeted antioxidants and cytotoxic agents Acta Pharm. Sinica B 2017; 7(1):106–115

12.   Madan RK and Levitt J. A review of toxicity from topical salicylic acid preparations. J Am Acad Dermatol. 2014 70(4), 788-92.

13.   Mancuso, C. and Santangelo, R. Invited Review. Ferulic acid: Pharmacological and toxicological aspects Food Chem. Toxicol. 2014, 65,185-195.

14.   Markovitsi, D. Gustavsson T. and I. Vaya, Fluorescence ofDNA duplexes: from model helices to natural DNA, J. Phys.Chem. Lett., 2010, 1, 3271–3276.

15.   McKenzie RL, Aucamp PJ, Bais AF, Björn LO, Ilyas M, Madronich S. Ozone depletion and climate change: impacts on UV radiation. Photochem Photobiol Sci 2011, 10, 182-198.

16.   Moreno, S.R.F.; Freitas, R.S.; Rocha, E.K.; Lima-Filho, G.L.; Bernardo-Filho, a.M. Protection of plasmid DNA by a Ginkgo biloba extract from the effects of stannous chloride and the action on the labeling of blood elements with technetium-99m. Braz. J. Med. Biol. Res. 2004, 37, 267-271.

17.   Patrick, M. H. Studies on thymine-derived UV photo-products in DNA-I. Formation and biological role of pyrimidine adducts in DNA, Photochem. Photobiol. 1977, 25, 357–372.

18.   Pei, K. Ou, J. Huanga J. and Ou S. p-Coumaric acid and its conjugates: dietary sources, pharmacokinetic properties and biological activities. J Sci Food Agric 2016; 96: 2952–2962

19.   Schuch, A.P. and Menck, C.F.M. The genotoxic effects of DNA lesions induced by artificial UV-radiation and sunlight. J. Photochem. Photobiol. B: Biol. 2010,  99,  111–116.

20.   Schuch, A.P.; Galhardo, R.S; Lima-Bessa, K.M.; Schuch, N.J. and Menck, C.F.M. Development of a DNA-dosimeter system for monitoring the effects of solar-ultraviolet radiation. Photochem. Photobiol. Sci. 2009, 8, 111–120.

21.   Schuch, A.P.; Garcia, C.C.M.; Makita, K.; Menck, C.F.M. DNA damage as a biological sensor for environmental sunlight. Photochem. Photobiol. Sci. 2013, 12, 1259-1272.

22.   Schuch, A.P; Moraes, M.C.S.; Yagura, T. and Menck, C.F.M. Highly Sensitive Biological Assay for Determining the Photoprotective Efficacy of Sunscreen. Environ. Sci. Technol. 2014, 48, 11584−11590

23.   Vernhes, M.; González-Pumariega, M.; Schuch, A.P.; Menck, C.F.M.; Sánchez-Lamar, A. El extracto acuoso de Phyllanthus orbicularis K protege al ADN plasmídico del daño inducido por las radiaciones ultravioletas. ARS Pharm. 2013, 54, 16–23.

Reviewer 2 Report

see attached

Author Response

Thanks for your suggestion about our paper tittled: Evaluation of genotoxic and DNA photo-protective activity of Bryothamnion triquetrum and Halimeda incrassata seaweeds extract. In order to respond the reviewer comments some modification were made in document. We are sending a summary with the changes.

The aim of this paper was to evaluate the genotoxic and antigenotoxic effects of two seaweeds aqueous extracts. Sunlight ultraviolet (UV) radiation wavelength could be divided into UVA (315-400 nm), UVB (280-315 nm) and UVC (100-280 nm), from less to the most energetic and mutagenic. Short wavelength UVC radiations are completely absorbed by the ozone layer, but there are important artificial sources commonly used in medicine and for cosmetic purposes (De Flora, 2013). Furthermore, stratospheric ozone layer depletion, together with climate change and global warming, predict an uncertain scenario for UV Earth’s incidence in the next decades (McKenzie et al., 2011), rising concerns about its biological consequences and motivating worldwide scientific efforts to develop better photoprotection strategies (González et al., 2011). In this sense to used UVC radiation for testing DNA damage is quite common in literature (Patrick 1997; Douki et al. 2002, 2013; Markovitsi et al. 2010; Banyasz et al. 2012). In a future UVB and UVA radiation could be evaluated as you suggested. However it was not our initial goal. T4-endo V recognizes CPD damages in DNA molecule and produce SSB, changing plasmid conformation from supercoiled to relaxed or nicked form. However in photoprotection studies, natural products could inactive T4-endo V enzyme and "false" results would be obtained. For this reason the enzyme activity was first tested. Different times were assayed. The first exposure time (0 min) means that the enzyme was put in contact with seaweeds extracts and immediately irradiated plasmid DNA was added. A version of Vidal et al 2001 protocol for preparing extracts was provided.

The changes proposed and other several English corrections were made. 

We are attaching you a modified manuscript

Thank you for your time

Round 2

Reviewer 1 Report

UVC as UV source was used in the 1970 ies to address mechanistic questions. Meanwhile a variety of light sources with more relevant parts of the solar spectrum are easily available.

I strongly recommdent to keep that in mind.

Author Response

Dear Reviewer,

The aim of this paper was to evaluate the genotoxic and antigenotoxic effects of two seaweeds aqueous extracts. In this sense to used UVC radiation for testing DNA damage is quite common in literature (Patrick 1997; Douki et al. 2002; Markovitsi et al. 2010; Banyasz et al. 2012). In a future UVB and UVA radiation could be evaluated as you suggested. However it was not our initial goal.

Banyasz, A. Douki, T. Improta, R. Gustavsson, T. Onidas, D. Vaya, I. Perron M. and D. Markovitsi. Electronic excited states responsible for dimer formationupon UV absorption directly by thymine strands: joint experimental and theoretical study, J. Am. Chem. Soc. 2012, 134, 14834–14845.

Douki, T. Vadesne-Bauer G. and Cadet J., Formation of 2′-deoxyuridine hydrates upon exposure of nucleosides togamma radiation and UVC-irradiation of isolated andcellular DNA, Photochem. Photobiol. Sci. 2002, 1, 565–569.

Markovitsi, D. Gustavsson T. and I. Vaya, Fluorescence ofDNA duplexes: from model helices to natural DNA, J. Phys.Chem. Lett., 2010, 1, 3271–3276.

Patrick, M. H. Studies on thymine-derived UV photo-products in DNA-I. Formation and biological role of pyrimidine adducts in DNA, Photochem. Photobiol. 1977, 25, 357–372.